# Moral Distress and Burnout in Neonatal Intensive Care Unit Healthcare Providers: A Cross-Sectional Study in Italy

**DOI:** 10.3390/ijerph19148526

**Published:** 2022-07-12

**Authors:** Sara Carletto, Maria Chiara Ariotti, Giulia Garelli, Ludovica Di Noto, Paola Berchialla, Francesca Malandrone, Roberta Guardione, Floriana Boarino, Maria Francesca Campagnoli, Patrizia Savant Levet, Enrico Bertino, Luca Ostacoli, Alessandra Coscia

**Affiliations:** 1Department of Neuroscience “Rita Levi Montalcini”, University of Torino, 10126 Turin, TO, Italy; sara.carletto@unito.it; 2Clinical Psychology Unit, A.O.U. City of Health and Science of Torino, 10126 Turin, TO, Italy; giulia.garelli460@edu.unito.it (G.G.); luca.ostacoli@unito.it (L.O.); 3Neonatal Intensive Care Unit of University of Torino, Sant’Anna Hospital, City of Health and Science, 10126 Turin, TO, Italy; mariachiara.ariotti@unito.it (M.C.A.); enrico.bertino@unito.it (E.B.); alessandra.coscia@unito.it (A.C.); 4Formerly at the School of Medicine, University of Torino, 10126 Turin, TO, Italy; ludovica.dinoto@outlook.it; 5Department of Clinical and Biological Sciences, University of Torino, 10043 Turin, TO, Italy; paola.berchialla@unito.it; 6Neonatal Care Unit, City of Health and Science University Hospital of Torino, 10126 Turin, TO, Italy; roberta.guardione@unito.it (R.G.); macampagnoli@cittadellasalute.to.it (M.F.C.); 7Neonatal Care Unit, Santa Croce Hospital ASL TO5, 10024 Moncalieri, TO, Italy; boarino.floriana@aslto5.piemonte.it; 8Neonatal Intensive Care Unit, Maria Vittoria Hospital, ASL Città di Torino, 10144 Turin, TO, Italy; patrizia.savantlevet@aslcittaditorino.it

**Keywords:** moral distress, burnout, NICU, healthcare workers, spirituality, existential orientation, religiousness

## Abstract

Moral distress (MD) in healthcare providers is widely recognized as a serious issue in critical care contexts. It has the potential to have negative impacts on both personal and professional wellbeing, the quality of care provided and staff turnover. The aim of this study was to investigate the relationship between MD and burnout among neonatal intensive care unit (NICU) healthcare professionals and identify the possible factors associated with its occurrence. Participants were asked to complete an online survey, which covered sociodemographic and professional information and included two self-report questionnaires (Italian Moral Distress Scale-Revised and Maslach Burnout Inventory). The sample comprised 115 healthcare providers (nurses and physiotherapists: 66.1%; physicians: 30.4%; healthcare assistants: 3.5%) working in four NICUs located within the province of Turin, Italy. The results revealed overall low levels of MD, with no significant differences between nurses/physiotherapists and physicians. Nurses/physiotherapists showed a statistically significant higher percentage of personal accomplishment burnout (32.9%) compared with physicians (8.6%; *p* = 0.012). MD was associated with the emotional exhaustion dimension of burnout. Spirituality and/or religiousness was shown to be a moderating variable. Further research is needed to deepen our understanding of the correlation between MD and burnout and the role of spirituality and/or religiousness as moderators.

## 1. Introduction

Moral distress (MD) concerns the distress associated with the moral integrity of a person. Its construct was first theorized in the 1980s by Jameton [1], and the first qualitative research on MD, conducted shortly afterwards by Wilkinson, contributed towards identifying the main situations in which MD arises and outlined its main consequences on the individual [2,3]. Since then, many authors have offered further elaborations of the original definition of MD in order to facilitate its application and identify the circumstances in which it may emerge [4]. Briefly, MD refers to the psychological, emotional and physiological suffering that arises when an individual’s actions contrast with his/her deepest ethical values, principles or commitments. A number of key necessary and/or sufficient conditions have been identified that explain the occurrence of this phenomenon, and they can be used to differentiate it from other forms of distress [4]. MD occurs when a person is required to perform a task perceived by the subject to violate their values [4,5,6]. This may arise due to perceived external constraints, which may involve characteristics of their working environment, be related to their specific profession, or be related to the quality of their unit/team work relationships [7,8,9].

MD can have many consequences and be experienced from different perspectives. It can have negative impacts on both personal and professional life spheres [10]. Consequences within the personal sphere may provoke emotional responses, manifesting as depression, anxiety, anger or frustration, physical disorders such as headaches or sleeping disorders [11,12,13], and psychological effects, which have an impact on the subject’s coping strategies and may bring about a loss of autonomy and the sense of disempowerment [1,2,9,14]. Within the professional sphere, MD can impact the quality of the work being performed and, as a consequence, the worker’s sense of job satisfaction [4,15,16,17]. Furthermore, MD may contribute to the decision of healthcare providers to leave their profession, and thus, raise staff turnover within a healthcare facility [3,18,19,20]. The relationship between MD and employee turnover is also influenced by other factors, such as burnout and the ethical climate of the organization [21,22,23].

Burnout is caused by excessive and prolonged stress, and it can be considered as a result of a disturbance of the equilibrium between the job demands individuals are exposed to and the resources they have at their disposal [24]. It is characterized by a state of emotional, physical and mental exhaustion, and includes elements of emotional exhaustion (EE), depersonalization (DP) and diminished personal accomplishment (PA) [25]. Several studies have confirmed an association between MD and burnout [18,26,27,28,29], with burnout widely considered to be an effect of MD [6,28,30,31], although the relationship between the two has yet to be fully elucidated. MD can lead to moral apathy, feelings of helplessness, lack of confidence, anxiety, frustration and anger, as well as to other situations that can further contribute towards the development of burnout [28,32,33,34].

Due to the nature of careers in healthcare, and the high commitment levels required to work in clinical situations, especially in contexts characterized by high stress levels, it is not surprising that MD has been largely studied and detected in medical care scenarios [15]. In fact, it is widely acknowledged that all healthcare professionals may experience MD, albeit in different forms and at different intensities [18,35,36,37,38].

Intensive care units (ICUs) are medical care environments in which healthcare professionals are highly susceptible to experiencing MD due to the highly vulnerable nature of their patients, for whom every decision and action taken is of utmost importance. Disagreements between colleagues within an ICU are also likely to be highly charged, exacerbated by the necessity for timeliness, the appropriateness of intervention choices made, the responsibility for making end-of-life decisions and the uncertainty of treatment outcomes [26,39,40]. This point is especially true for neonatal intensive care units (NICUs), in which the medical staff are responsible for caring for premature and critically ill neonates, treatment choices, interacting with the parents and making treatment and care decisions for babies in life-threatening conditions [18,35,36,41,42]. MD experienced by healthcare workers in the NICUs can negatively affect their ability to provide the best quality of care and interact in the best possible way with the patients’ families [18].

Studies conducted in the context of NICUs have highlighted some demographic characteristics associated with the probability of experiencing MD. Although it is important to recognize that the risk of MD is a reality for all practitioners in the neonatal ward, it has been especially linked with nursing professions. This is probably a consequence of their perceived lack of power in making decisions and difficulties experienced in nurse–physician interprofessional relationships and teamwork [39]. Previous studies have also identified the number of years working in the NICU as both a risk factor and a protective factor: on the one hand, the more that a healthcare professional is exposed to morally distressing situations, the more likely they are to accumulate unresolved distress (the so-called “crescendo effect”) [6]; on the other hand, the longer a healthcare provider remains in a profession, the more likely they are to feel confident about their performances and the less likely they may be to quit their position [10,36,43].

Other studies have shown a possible correlation between self-reported levels of religiousness and spirituality in healthcare professionals and the experiencing of MD, but the results published on this aspect to date remain unclear and controversial [8,10,16,30,34,35,44,45]. Lastly, various studies have emphasized the need to investigate the occurrence of MD and burnout in NICU workers while simultaneously investigating the contribution of other factors in order to understand the nature of their interaction better [18,32,46].

In the Italian context, a limited number of studies have explored MD among healthcare professionals [47,48,49,50,51,52,53,54], but only one study involved those working in NICUs [55]. The study showed low-to-moderate MD levels, but the sample consisted of nurses only and the relationship between MD and burnout was not investigated. Therefore, the aim of the present study was to investigate MD and burnout levels in NICU healthcare providers and to identify the factors associated with its occurrence.

## 2. Materials and Methods

### 2.1. Design and Participants

The sample of this cross-sectional study was composed of healthcare providers working in any one of the four NICUs operating within the province of Turin, Italy (two NICUs in Sant’Anna Hospital, University Hospital “Città della Salute e della Scienza di Torino”, Turin; the NICU in Maria Vittoria Hospital, ASL Città di Torino, Turin; and the NICU in Santa Croce Hospital, ASL TO5, Moncalieri). Participants were eligible for inclusion if they worked as physicians, nurses, physiotherapists or healthcare assistants in one of the aforementioned NICUs. Those who did not provide their consent were excluded from the study. Healthcare providers were recruited from December 2021 to February 2022 through an email sent to their institutional email address which invited them to participate in the electronic survey set on the Google Forms platform. The study was conducted in accordance with the Declaration of Helsinki, and the research protocol was approved by the Ethics Committee of the University of Turin. Informed consent was obtained by asking all participants to click a button at the beginning of the online survey consenting to their participation. Participation was voluntary and anonymous, and participants received no compensation.

### 2.2. Study Measures

The online survey took about 15 min to complete and was composed of two sections. The first included questions about sociodemographic factors (age, educational level, living status, children), professional information (profession, years of professional experience, years of professional experience in the NICU, full- or part-time contract, having left work for reasons unrelated to MD), and data about their religious and spiritual orientation. Regarding the latter, as religion and spirituality are not mutually exclusive but possess overlapping as well as distinct elements [56], we asked participants to define which of the following categories they identified with most: (i) neither religious nor spiritual; (ii) spiritual but not religious; (iii) religious but not spiritual; (iv) both spiritual and religious.

The second section comprised two validated self-report questionnaires:The Italian Moral Distress Scale-Revised (MDS-R) [49], a 14-item self-report instrument to assess moral distress among critical care clinicians. Each item was scored on two 5-point Likert scales from 0 to 4, assessing the intensity and frequency of moral distress. For each item, a composite score was computed by multiplying the frequency by the intensity scores, generating an overall score ranging from 0 to 16, with higher scores indicating higher levels of moral distress.The Maslach Burnout Inventory (MBI) [57], a 22-item self-report questionnaire assessing burnout, divided into three domains: emotional exhaustion (EE; 9 items), depersonalization (DP; 5 items), and personal accomplishment (PA; 8 items). The frequency of symptoms within each of these domains was scored on a 7-point Likert scale from 0 (never) to 6 (every day), and summarized as continuous variables on the basis of a composite score (EE 0–54, DP 0–30, and PA 0–48). Each scale was devised such that higher scores indicate more of each construct. Higher scores on the EE and DP subscales indicate a higher degree of burnout symptoms, whereas lower scores on the PA subscale indicate a higher burnout burden. The Italian validation of the questionnaire established the following ranges for high, medium or low levels of each construct in healthcare providers: high: EE ≥ 24, DP ≥ 9, PA ≥ 37; medium: EE = 15–23, DP = 4–8, PA = 30–36; low: EE ≤ 14, DP ≤ 3, PA ≤ 29 [58].

### 2.3. Statistical Analysis

Continuous variables were expressed as the median plus interquartile range (IQR), and groups were compared using the Mann–Whitney U test. Categorical variables were summarized as counts and percentages, and the χ2 test or Fisher’s test was performed, as appropriate, to test for differences between groups. A univariate analysis based on a linear regression model was carried out to identify associations between sociodemographic variables and MDS-R, MBI-EE, MBI-DP and MBI-PA. Coefficients and 95% confidence intervals (95% CIs) were reported. A final multivariate model was developed based on statistical selection procedures. Model selection was performed using an automatic approach based on the Akaike information criterion (AIC) method. Given the large number of covariates, a genetic algorithm was employed to explore the candidate set of models. The significance level was set at *p* < 0.05. All statistical analyses were performed using R version 4.1.2 (R Company, https://www.r-project.org).

## 3. Results

### 3.1. Subject Characteristics

A total of 197 healthcare providers were asked to participate in the online survey; 115 questionnaires were collected, giving a response rate of 58.4%. Sample characteristics, summarized as percentages, are shown in Table 1. Nurses and physiotherapists (grouped together into a single category) made up 66.1% of the sample, and thus constituted the majority of participants; 30.4% were physicians, and only 3.5% were healthcare assistants. Just over half of the sample (55.7%) reported to have been working in the NICU for more than 10 years.

### 3.2. Moral Distress and Burnout Levels

MDS-R total scores ranged from 0.2 to 13 for nurses/physiotherapists, with a median score of 4.36 (IQR: 2.39–5.64); for physicians it ranged from 1.7 to 10, with a median score of 3.36 (IQR: 2.68–4.11) (Table 2). No significant differences between groups were observed. Table 2 also reports the total scores for the three burnout dimensions, MBI-EE, MBI-DP and MBI-PA, and the percentage of respondents with scores of MBI-EE ≥ 24, MBI-DP ≥ 9 and MBI-PA ≤ 29, which indicate burnout as a dichotomous variable. A significantly higher percentage of nurses/physiotherapists were classified as experiencing burnout on the Personal Accomplishment (PA) scale (32.9%) compared with physicians (8.6%) (*p* = 0.012).

### 3.3. Associations between Other Variables, Moral Distress and Burnout

Initial univariate analyses of the relationship between demographic variables and the MDS-R score indicated no statistically significant association with the professional category, other sociodemographic variables, professional experience or spirituality with moral distress or burnout (Table 3).

Using statistical selection techniques, a linear regression model was implemented to explore in more detail the presence of potential associations between respondent variables and MDS-R scores. A genetic algorithm was applied to find the best model in terms of AIC, considering all the explanatory variables in the univariate analysis along with the MBI scale score as a potential mediator. The resulting model showed MBI-EE to be significantly associated with MDS-R: for each unit increase in MBI-EE, the MDS-R score also increased by a mean of 0.13 (95%CI 0.06–0.21, *p* = 0.001) in individuals with a comparable spirituality rating and MBI-DP (Table 4). In addition, the increase was statistically smaller among more religious and/or spiritual individuals compared with individuals reporting no religious or spiritual inclination (Table 4).

Figure 1 depicts the correlation between MDS-R and the MBI scale. The correlations between MDS-R and MBI-EE, MBI-DP and MBI-PA are 0.3 (*p* = 0.001), 0.29 (*p* = 0.002) and –0.18 (*p* = 0.051), respectively.

## 4. Discussion

To the best of our knowledge, this study is the first to investigate moral distress and burnout in healthcare providers working in Italian NICUs. As highlighted in previous research, healthcare professionals working in NICU settings may be at a higher risk of developing MD than in other hospital departments, which may in turn have negative effects on both personal and professional wellbeing, as well as significant repercussions on patient care [35,41].

This study sample reported overall low levels of MD in healthcare professionals from the four NICUs located in Northern Italy. MDS-R scores were slightly lower in physicians than in nurses/physiotherapists, although the difference was not significant. Although this result contrasts with that of some studies which found higher MD levels in nurses than in other professional categories [15,18,26,43,53,54,59], it is in line with others which instead found no differences in MD levels between physicians and non-physicians [36,49,60]. The differences between the various studies may be due to the different social and cultural characteristics of the study samples and the different organizational contexts. Very few studies exist in the literature which have investigated MD in all the professional categories working in NICUs; thus, further studies are needed to better clarify the presence of possible differences.

The prevalence of the three underlying burnout dimensions in our sample (high EE: 36%; high DP: 19.8%; low PA: 25.2%) was similar to that reported in a recent meta-analysis on pediatric nurses (high EE: 31%; high DP; 21%; low PA: 39%) [61]. The comparison of professional categories revealed no significant difference in the percentage of nurses/physiotherapists experiencing high EE compared with physicians (28% vs. 12%, *p* = 0.962). There was a non-significant trend for high DP to be more prevalent in nurses/physiotherapists compared with physicians (19% vs. 3%, *p* = 0.078), and a very significant trend for low PA in the former vs. the latter (32.9% vs. 8.6%, *p* = 0.012). Previous studies investigating burnout dimensions in pediatric healthcare professionals showed its development to be multidimensional and largely influenced by personal, professional and organizational/contextual factors [62]. Indeed, the exact role of the professional category as a factor associated with burnout risk remains unclear in the existing literature, with inconsistent results across studies. Some studies showed that nursing professionals and respiratory therapists have significantly higher levels of burnout than physicians [63,64,65], whereas other studies showed higher levels of burnout in physicians [66,67]. A low PA level indicates the presence of significant feelings of perceived ineffectiveness and frustration regarding work-related achievements. In particular, low PA seems to be associated with poor task significance, poor task clarity, low decision latitude and the lack of perceived social support [68]. With this in mind, we might hypothesize that nurses/physiotherapists would be more likely to experience these types of frustrations compared with physicians, given their job roles and limitations when looking after NICU patients. Although burnout is increasingly recognized as a significant problem among healthcare professionals working in intensive care units [69], few studies have focused on pediatric intensive care, and even fewer on the NICU context [70]. Considering the significant impact burnout can have at multiple levels (personal wellbeing, the quality of care offered to patients and families, the organization), further studies are needed to investigate the severity and prevalence of this syndrome in healthcare providers working in NICUs [71].

The results of our study revealed a correlation between MD and burnout levels, corroborating the results of numerous other studies from an array of healthcare contexts [18,28,30,32,46,51,72,73,74,75,76], especially in relation to EE and DP [26,47,77]. It is widely acknowledged that healthcare providers working in NICUs may be exposed to various ethical problems and stressors related to the clinical situations of the patients and their families (e.g., critically ill or dying patients; dealing with the families’ perception of certain treatments as being futile) as well as the relational dynamics of their work group/interdisciplinary medical team (e.g., poor communication) and the organizational context (e.g., lack of resources), which can act as determinants for both MD and burnout [47]. Although early studies described burnout as a consequence of prolonged exposure to MD [6,28,30], more recent studies have highlighted the relationship between MD and burnout to be part of a complex interplay, in which other mediating and moderating factors also play key roles [46,72,78].

The present study revealed a significant association between MD and EE moderated by spirituality and/or religiousness. In fact, the association between EE and MD was significantly weaker among religious and/or spiritual individuals as compared with those who considered themselves non-religious and non-spiritual. Individual spiritual and religious beliefs can have an impact on the perception of MD [45]. One study showed that experiences of MD can negatively influence spiritual well-being [8]. Moreover, studies conducted in non-Western countries, in which religion plays a more prominent role in people’s sociocultural life, evidenced that the conflict between professional values and religious beliefs may contribute to the development of MD [75,79]. On the other hand, being spiritual and/or religious may act as a protective factor for both MD and the EE dimension of burnout [16,30,73,75,80]; one explanation is that it may decrease perceived suffering and dissatisfaction related to MD [80]. Spiritual and/or religious resources may provide an individual with coping skills [10,81], helping the subject to manage feelings of helplessness, reducing feelings of worthlessness, and promoting a personal sense of meaning [45,80] whilst confronting morally distressing situations. Very few studies have considered the role of religiosity/spirituality as a factor related to MD in the context of NICUs. In the study conducted by Catlin et al. [45], healthcare workers considered spirituality and religion to be a central element contributing to patient-centered care, and most of them (83%) declared to pray in private for the NICUs’ newborns. Praying was also recognized as a source of relief and refuge when dealing with the most distressing patient cases. Cavaliere and colleagues [16] found that NICU nurses who were not spiritual had significantly higher levels of MD than those who were “very” or “somewhat” spiritual. Finally, Kukora and Boss [82] pointed out the need for NICU healthcare professionals to take the religious and spiritual values of the parents into consideration when embarking on shared decision-making processes related to a newborn’s care. To this end, the need to incorporate spirituality into training courses for healthcare providers working in NICUs is crucial, not only to mitigate their emotional burden, and thus reducing levels of MD and burnout, but also to improve the quality of care offered in a family-centered care perspective [83]. As very few studies have been conducted on these issues thus far, further research is needed to deepen our understanding of the role of spirituality and religiousness as moderators of the relationship between burnout and MD.

The main strengths of this study are that it investigated MD together with all three dimensions of burnout, and revealed a significant role of spirituality/religiosity as one of the potential moderating effects of the MD–burnout relationship. The study also has a number of limitations. First, due it its cross-sectional design, only associations between the variables studied can be detected, precluding definitive conclusions regarding the direction of causality. Second, the generalizability of our findings is limited by the respondents’ self-selection of responses, the lack of data on non-responders, the limited number of NICU healthcare providers participating in the study and the restricted geographical area in which these four NICUs are located. Third, data regarding gender were not collected in order to ensure the anonymity of the participants. Future multicenter studies with larger sample sizes are needed to deepen our understanding of MD and burnout among NICU healthcare providers, particularly in the Italian context, as the results would help identify strategies to alleviate both MD and burnout.

Future studies might also consider using a pediatric-specific instrument for MD (see Grasso et al. [53]), pending the development and validation of an NICU-specific version. In fact, the NICU context is peculiar due to the heterogeneity of the newborn populations and their clinical conditions (e.g., pre-term, hypoxic-ischemic encephalopathy, congenital malformations, etc.) and, therefore, the heterogeneity of the associated ethical issues. It is also crucial to consider other “individual” factors, such as those related to the emotional intelligence [84] and work–life balance of the professional [78], “contextual” factors (e.g., family perspectives) [85], and “organizational” factors. Such an approach would help to identify those individuals most at risk of developing MD and burnout [23,86] and to investigate the complex interplay between these variables. Potentially morally challenging situations are an inevitable component of the NICU healthcare providers’ experience [18], and can be considered a potential source of growth and self-improvement if exploited correctly [41]. At the same time, if MD remains undetected and without any appropriate interventions, it can have significant repercussions on the wellbeing of healthcare providers, on the work climate of the interdisciplinary medical team and on the quality of the care provided to patients and families [41,86].

## 5. Conclusions

In conclusion, this study revealed an association between MD levels in NICU healthcare providers and levels of burnout, in particular the EE dimension; this relationship was also shown to be moderated by spirituality/religiousness. Further research is needed to fully explore the role of religious beliefs and spiritual well-being on MD and burnout whilst bearing in mind the possible influence of different sociocultural components.

## Figures and Tables

**Figure 1 ijerph-19-08526-f001:**
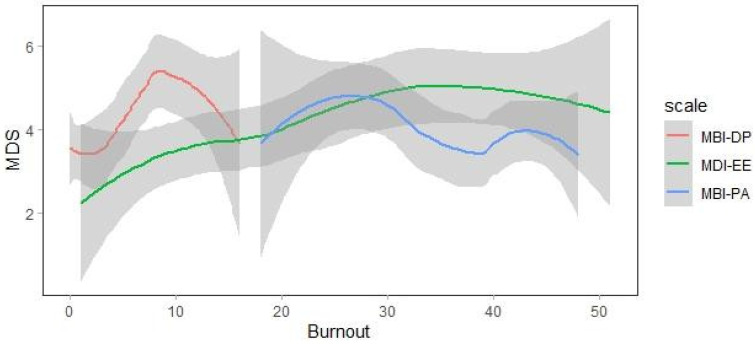
Correlation between MDS-R and MBI-EE, MBI-DP and MBI-PA scores.

**Table 1 ijerph-19-08526-t001:** Sample characteristics.

	Overall
	*N* = 115
Professional category (%)	
Nurses/physiotherapists	76 (66.1)
Physicians	35 (30.4)
Healthcare assistants	4 (3.5)
Age group (%)	
21–25	8 (7.0)
26–30	9 (7.8)
31–35	18 (15.7)
36–40	17 (14.8)
41–45	12 (10.4)
46–50	20 (17.4)
51–55	16 (13.9)
56–60	11 (9.6)
>60	4 (3.5)
Housing condition = living with other people (%)	95 (82.6)
Children = yes (%)	72 (62.6)
Religious (%)	
Atheist	17 (14.8)
Non-practicing believer	73 (63.5)
Very religious	25 (21.7)
Existential orientation (%)	
Neither religious nor spiritual	28 (24.3)
Spiritual but not religious	42 (36.5)
Religious but not spiritual	10 (8.7)
Both spiritual and religious	35 (30.4)
Employment contract = full time (%)	109 (94.8)
Years working as healthcare professional (%)	
<5	15 (13.0)
5–10	23 (20.0)
11–20	29 (25.2)
>20	48 (41.7)
Years working in the NICU (%)	
<5	34 (29.6)
5–10	17 (14.8)
11–20	30 (26.1)
>20	34 (29.6)
Educational level (%)	
Professional qualification	23 (20.0)
Bachelor’s degree	52 (45.2)
Master’s degree	6 (5.2)
Single-cycle degree	34 (29.6)
Left job for reasons not related to MD = yes (%)	14 (12.2)

**Table 2 ijerph-19-08526-t002:** Moral Distress and MBI scale scores for nurses/physiotherapists and physicians.

	Nurses/Physiotherapists	Physicians	*p*
	*N* = 76	*N* = 35	
MDS-R, median [IQR]	4.36 [2.39, 5.64]	3.36 [2.68, 4.11]	0.257
MBI-EE, median [IQR]	19.00 [12.00, 30.75]	19.00 [11.50, 33.50]	0.906
MBI-DP, median [IQR]	4.00 [1.00, 8.25]	3.00 [1.00, 6.50]	0.661
MBI-PA, median [IQR]	35.00 [28.75, 42.25]	38.00 [34.00, 41.00]	0.265
MBI-EE ≥ 24, *N* (%)	28 (36.8)	12 (34.3)	0.962
MBI-DP ≥ 9, *N* (%)	19 (25.0)	3 (8.6)	0.078
MBI-PA ≤ 29, *N* (%)	25 (32.9)	3 (8.6)	0.012

**Table 3 ijerph-19-08526-t003:** Associations between sociodemographic variables, professional experience and spirituality with MDS and MBI scales.

	MDS-R	MBI-EE	MBI-DP	MBI-PA
	B(95%CI)	B(95%CI)	B(95%CI)	B(95%CI)
Professional category				
Nurses/physiotherapists	-	-	-	-
Physicians	–0.51 (–1.47–0.45)	–0.38 (–5.48–4.72)	–0.90 (–2.70–0.89)	1.87 (–1.23–4.97)
Age				
Over 40	-	-	-	-
Under 40	0.48 (–0.41–1.38)	–3.35 (–8.05–1.36)	1.08 (–0.59–2.75)	–1.89 (–4.77–0.99)
Housing condition				
Living alone	-	-	-	-
Living with other people	–0.82 (–1.98–0.34)	–0.51 (–6.67–5.66)	–1.74 (–3.89–0.42)	1.64 (–2.12–5.40)
Children				
No	-	-	-	-
Yes	–0.61 (–1.53–0.31)	1.05 (–3.81 to 5.91)	–0.72 (–2.44–1.00)	–0.24 (–3.22–2.74)
Religious				
Atheist	-	-	-	-
Non-practicing believer	0.39 (–0.89–1.68)	–1.23 (–7.96–5.50)	–0.67 (–3.04–1.70)	–1.04 (–5.15–3.07)
Very religious	0.20 (–1.31–1.70)	2.60 (–5.29–10.49)	0.99 (–1.79–3.77)	–3.21 (–8.03–1.61)
Existential orientation				
Neither religious nor spiritual	-	-	-	-
Spiritual but not religious	–0.01 (–1.19–1.17)	3.68 (–2.51–9.88)	–0.42 (–2.62–1.78)	1.29 (–2.51–5.08)
Religious but not spiritual	0.63 (–1.20–2.47)	1.30 (–8.33–10.92)	–0.33 (–3.75–3.08)	–2.15 (–8.04–3.74)
Both spiritual and religious	–0.20 (–1.43–1.02)	3.40 (–3.05–9.84)	0.63 (–1.66–2.92)	0.17 (–3.77–4.12)
Contract of employment				
Part time	-	-	-	-
Full time	–1.82 (–3.78–0.14)	–3.60 (–14.05–6.86)	0.54 (–3.17–4.25)	1.53 (–4.88–7.93)
Years as healthcare professional				
<5	-	-	-	-
5–10	0.43 (–1.15–2.00)	6.82 (–1.34–14.98)	–0.49 (–3.44–2.46)	–3.01 (–8.08–2.05)
11–20	–0.33 (–1.85–1.20)	2.60 (–5.32–10.52)	–1.01 (–3.87–1.84)	–2.50 (–7.41–2.42)
>20	–0.35 (–1.76–1.06)	7.21 (–0.10–14.52)	–1.23 (–3.87–1.41)	–0.86 (–5.40–3.68)
Years working in NICU				
<5	-	-	-	-
5–10	0.77 (–0.67–2.22)	–0.09 (–7.62–7.43)	2.62 (0.01–5.24) *	–2.16 (–6.82–2.51)
11–20	–0.48 (–1.68–0.71)	–3.32 (–9.57–2.92)	–1.40 (–3.57–0.76)	–0.31 (–4.18–3.57)
>20	0.12 (–1.05–1.29)	4.00 (–2.10–10.09)	–0.88 (–2.99–1.24)	1.46 (–2.32–5.24)
Educational level				
Professional qualification	-	-	-	-
Bachelor’s degree	–0.60 (–1.82–0.62)	–6.15 (–12.68–0.38)	–0.46 (–2.79–1.87)	–0.20 (–4.05–3.66)
Master’s degree	1.78 (–0.54–4.10)	–3.70 (–16.10–8.70)	0.95 (–3.47–5.37)	–11.15 (–18.48––3.82) **
Single-cycle degree	–0.91 (–2.21–0.40)	–4.64 (–11.62–2.35)	–1.31 (–3.80–1.18)	0.62 (–3.51–4.75)
Left job for reasons not related to MD				
No	-	-	-	-
Yes	–0.32 (–1.67–1.03)	–1.56 (–8.69–5.57)	1.68 (–0.83–4.18)	2.68 (–1.66–7.01)

* *p* < 0.05; ** *p* < 0.01.

**Table 4 ijerph-19-08526-t004:** Multivariate regression model with MDS-R as the dependent variable.

	Coefficient	95%CI	*p*-Value
MBI-EE score	0.13	0.06 to 0.21	0.001
Existential orientation			
Neither religious nor spiritual	-	-	-
Spiritual but not religious	1.77	–0.45 to 4	0.117
Religious but not spiritual	4	1.04 to 6.96	0.009
Both spiritual and religious	1.69	–0.75 to 4.13	0.172
MBI-DP ≥ 9	0.79	–0.33 to 1.91	0.165
MBI-EE score × spiritual but not religious	–0.1	–0.2 to –0.005	0.041
MBI-EE score × religious but not spiritual	–0.17	–0.29 to –0.05	0.006
MBI-EE score × both spiritual and religious	–0.10	–0.21 to –0.001	0.049

## Data Availability

The data that support the findings of this study are available from the corresponding author (S.C.) upon reasonable request.

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
