# Peer review of "Moral Distress and Burnout in Neonatal Intensive Care Unit Healthcare Providers: A Cross-Sectional Study in Italy"

_ijerph, 2022, doi:10.3390/ijerph19148526_

Round 1

Reviewer 1 Report

Dear authors,

I found your paper very interesting, with particular reference to the role of religion and spirituality as a protective factor against MD.

Please find some recommendations which I think would improve the value of this paper, first of all by considering similar, recent works published on the topic in the Italian context.

Extensive language revision by a mother-tongue is recommended to improve fluency.

L. 28 Replace "scenario" with "contexts".

L. 61-62 lack of knowledge about a specific situation in which a subject is required to act" is more properly moral uncertainty, which along with moral distress and moral dilemma is a sub-dimension of moral stress. The following words related to the external constraints are instead more adequate to define moral distress and I suggest to synthesize the entire period (From "MD occurs" to "work relationships" only utilizing this definition.
L. 67 I recommend to put a citation which works well with this proposition, that is Villa et al., 2021 (Moral Distress in Community and Hospital Settings for the Care of Elderly People).

LL. 75-77 I suggest to clarify here and elsewhere the relationship between moral distress and burnout; it is not clear to me what is the cause and what is the effect, while moral distress is generally meant as a cause of burnout. If I can help you see for instance Wagner, 2015 (Moral Distress as a Contributor to Nurse Burnout), Saarnio et al. 2012 (Stress of conscience among staff caring for older persons in Finland) and Pennestrì et al. 2022 (Non c'è cura dei pazienti senza cura degli operatori). I have seen that this relation is clarified later on, LL 296-299. It would be useful for the reader to write first that this relation is unclear, before than LL 125-126; then to write in the discussion that MD-burnout is a complex interplay (although I still believe that MD is cause of burnout much more than the opposite).

LL. 91-92 Please remove at least one of the redunand "and" conjunctions.

L. 107 ("the so-called crescendo effect)

L. 109 Villa et al. 2021 is a useful citation here too.

L. 112 same as before

L. 128 Giannetta et al. 2021 (Levels of moral distress among health care professionals working in hospitals and community care: a cross sectional study) also have explored MD among HCPs in Italy.

L. 155 I find very interesting and appropriate the idea to include religion and spiritual orientation in the data. Faith is a protective factor against moral distress and burnout. That's why in Table 1 I thinks it's fundamental to dedicate a couple of words about what is meant with "religious and spiritual", "spiritual but not religious" and so on.

LL. 203-206 It would be interesting to discuss these findings in light of the findings of Giannetta et al. 2021, where nurses resulted the most exposed HCPs to MD followed by phisicians, and physical therapists the least. For instance considering potential reasons behind this difference.

L. 312 I recommend to add Villa et al. 2021 here

LL. 324-330 Very useful lines.

Reviewer 2 Report

I would like to thank you for giving me this opportunity to evaluate this scientific paper, which focuses moral distress and burnout among healthcare professional working neonatal intensive care unit (NICU)

This manuscript reports new findings and is theoretically based on the current literature.

 - The manuscript is within the journal's scope.

- This study was well designed, executed, and presented.

- Figures and tables are well presented

- The conclusion is consistent with the evidence presented

- The discussion is relevant

- References are up to date and relevant, but some if the references are not in the format required for the journal (21, 31, 38, 39,43).

In Material and Methods I would like to know when this study has been done.
